# Safflower Flavonoid 3′5′Hydroxylase Promotes Methyl Jasmonate-Induced Anthocyanin Accumulation in Transgenic Plants

**DOI:** 10.3390/molecules28073205

**Published:** 2023-04-04

**Authors:** Xinyue Zhang, Naveed Ahmad, Qingyu Zhang, Abdul Wakeel Umar, Nan Wang, Xu Zhao, Kang Zhou, Na Yao, Xiuming Liu

**Affiliations:** 1Engineering Research Center of the Chinese Ministry of Education for Bioreactor and Pharmaceutical Development, College of Life Sciences, Jilin Agricultural University, Changchun 130118, China; 2Joint Center for Single Cell Biology, Shanghai Collaborative Innovation Center of Agri-Seeds, School of Agriculture and Biology, Shanghai Jiao Tong University, Shanghai 200240, China; 3BNU-HKUST Laboratory of Green Innovation, Advanced Institute of Natural Sciences, Beijing Normal University, Zhuhai 519088, China; 4Jilin Province Institute of Product Quality Supervision and Inspection, Changchun 130022, China; 5Jilin Province Science and Technology Information Research Institute, Shenzhen Street 940, Changchun 130033, China

**Keywords:** safflower, flavonoid, *F3′5′H*, methyl jasmonate, anthocyanin, VIGS

## Abstract

Flavonoids are the most abundant class of secondary metabolites that are ubiquitously involved in plant development and resistance to biotic and abiotic stresses. Flavonoid biosynthesis involves multiple channels of orchestrated molecular regulatory factors. Methyl jasmonate (MeJA) has been demonstrated to enhance flavonoid accumulation in numerous plant species; however, the underlying molecular mechanism of MeJA-induced flavonoid biosynthesis in safflower is still not evident. In the present study, we revealed the underlying molecular basis of a putative *F3′5′H* gene from safflower imparting MeJA-induced flavonoid accumulation in transgenic plants. The constitutive expression of the *CtF3′5′H1* gene was validated at different flowering stages, indicating their diverse transcriptional regulation through flower development in safflower. Similarly, the *CtF3′5′H1*-overexpressed Arabidopsis plants exhibit a higher expression level, with significantly increased anthocyanins and flavonoid content, but less proanthocyanidins than wild-type plants. In addition, transgenic plants treated with exogenous MeJA revealed the up-regulation of *CtF3′5′H1* expression over different time points with significantly enhanced anthocyanin and flavonoid content as confirmed by HPLC analysis. Moreover, *CtF3′5′H1*- overexpressed Arabidopsis plants under methyl violet and UV-B irradiation also indicated significant increase in the expression level of *CtF3′5′H1* with improved anthocyanin and flavonoid content, respectively. Noticeably, the virus-induced gene silencing (VIGS) assay of *CtF3′5′H1* in safflower leaves also confirmed reduced anthocyanin accumulation. However, the *CtF3′5′H1* suppression in safflower leaves under MeJA elicitation demonstrated significant increase in total flavonoid content. Together, our findings confirmed that *CtF3′5′H1* is likely mediating methyl jasmonate-induced flavonoid biosynthesis in transgenic plants via enhanced anthocyanin accumulation.

## 1. Introduction

Safflower (*Carthamus tinctorius* L.), an annual herb, contains dry tubular flowers with a special aroma and a slightly bitter taste. Their floral parts consist of 20–30% yellow pigment and 0.3–0.6% red pigment, and therefore safflower yellow pigment is the main active component of safflower [1,2]). Safflower yellow pigment is a water-soluble flavonoid; it has many pharmacological effects, including anti-diabetes, treatment of brain injury, and anti-cancer properties [3,4]. In China, safflower is often used in the treatment of cardiovascular and cerebrovascular diseases, promoting blood circulation and removing blood stasis and relieving joint pain [5]. In addition, safflower yellow pigment and other classes of flavonoids are known to inhibit bacterial growth by affecting the transcriptional and metabolic pathways in the host, and hence are widely used in anti-inflammatory and antibacterial agents [6]. At present, the study of flavonoids has entered the latest progress in microbial host selection and genetic coding biosensors [7]. Hence safflower demonstrates significant genetic variation for genome-wide association analyses of gene families linked to flavonoid biosynthesis, and it is expected that the discovery of safflower genome resources would further speed up the pace of biological evolution.

Flavonoids are the most common polyphenolic compounds containing at least one aromatic ring covalently linked with one or more hydroxyl groups. In plants, the flavonoid biosynthesis pathway is well-understood and includes several enzymatic stages that lead to several flavonoid classes [8,9]. Flavonoid 3′-hydroxylase (*F3′H*) and flavonoid 3′5′-hydroxylase (*F3′5′H*) are generally known to catalyze the hydroxylation of the B ring leading to the biosynthesis of hydroxylated flavonoids [10]. In the past, several studies have demonstrated the importance of *F3′5′H* encoding genes in a variety of plant species such as tea [11], potato [12], pea [13], and tomato [14]. A wide variety of flavonoid substrates, including flavanones, dihydroflavonols, flavonols, and flavones, could be hydroxylated by heterologously produced F3′5′Hs from tomato [14], tea [15], and grapevine [16]. However, the substrate naringenin appears to be the best substrate among others. The heterologous expression of *F3′5′Hs* from tomato, tea, and grapevine have been shown to hydroxylate the C3′ and/or C5′ positions of a broad range of flavonoid substrates, including flavanones, dihydroflavonols, flavonols, and/or flavones, with naringenin generally being the optimum substrate.

Flavonoid hydroxylases have attracted a lot of attention since their hydroxylation patterns have a significant influence on flower color. In addition to anthocyanins, the hydroxylation pattern of flavonols and proanthocyanidins is also regulated by the activity of F3′H and an F3′5′H, resulting in flavonols and Phenylalanine with various biochemical properties and antioxidant properties. It was shown that the *CYP75Hf1* and *CtCYP82G24* cDNA from *Pterolophia. Hybrida* and *C. tinctorius* [17,18], and the *F3′5′H* gene from *Carthamus. roseus* [19] were able to hydroxylate flavones, flavanones, dihydroflavonols, and flavonols as well as catalyze 3′- and 3′5′-hydroxylation. Similarly, a potato *F3′5′H* has been shown to alter anthocyanins from pelargonidin to petunidin derivatives, which resulted in the purple-skinned potato cultivar [12]. Prior studies also demonstrated that the suppression of the *F3′5′H* gene in transgenic potato tubers reduced anthocyanin production and resulted in a 100-fold spike in kaempferol concentration [20]. Studies also demonstrated that the Pea *F3′5′H* mutants lack the capacity to produce delphinidin and petunidin, which are the primary pigments in the wild-type pea flowers [13]. The 3′,4′,5′-hydroxylated anthocyanin was accumulated in transgenic Petunia overexpressed with *VvF3′5′H1*, resulting in a shift from kaempferol to quercetin in comparison to wild-type plants).

Jasmonates (JAs), also known as oxygenated fatty acids (oxylipins), are the type of plant hormones that are synthesized via octadecanoid/hexadecanoid pathways [21]. Studies have shown that widely known derivatives of jasmonate including MeJA and JA-Ile are able regulate pollen development [22], plant responses to biotic and abiotic stresses [23], and plant senescence [24]. The mechanism involving jasmonate-induced anthocyanin accumulation is well-studied in the model plant Arabidopsis as well as in other plants [25,26]. Prior studies have shown that the 26S proteasome pathway degrades JAZ proteins after they have been recruited to the SCFCOl1 complex by COl1 for proteasomal degradation upon the reception of the jasmonate signal [27]. This results in the production of the MBW complex containing MYB and bHLH transcription factors, which thereby activates the transcriptional activation of the core structural genes of the flavonoid biosynthesis pathway [28,29,30,31]. However, the core molecular mechanism that underlies how JA regulates flavonoid biosynthesis via anthocyanin accumulation has largely remained elusive.

Considering the genomic diversity and multicolor flowering system of safflower, it is essential to focus on genes/transcription factors that control the biosynthesis floral pigments as well as other specialized metabolites. There is, however, a quite limited amount of information available on the regulation of methyl jasmonate-induced flavonoid biosynthesis in safflower. In this study, we identified a putative *F3′5′H1* gene from safflower, likely regulating methyl jasmonate-induced flavonoid biosynthesis via promoting anthocyanin accumulation in transgenic Arabidopsis and safflower. This study will lay a foundation for understanding the putative role of flavonoid hydroxylases in methyl jasmonate induced regulation of specialized metabolites in plants.

## 2. Results

### 2.1. Identification, Physicochemical Properties, and Phylogenetic Analysis

In order to identify *F3′5′H* encoding genes in safflower, we performed an electronic search using the Pfam ID PF00067 against the safflower genome. According to the information of genome sequencing, two candidate sequences of safflower *CtF3′5′H* were identified and further subject to Basic Local Alignment Search Tool (BLAST) for similarity searches. The index of the grand average of hydropathicity (GRAVY) showed that these two proteins were likely stable with a stability index in the range of 33.27, aliphatic index of 28.63, and the pI 5.01, respectively. A phylogenetic tree was constructed to investigate the evolutionary relationship between F3′5′H sequences from safflower and other plant species. The phylogenetic tree contained a total of 73 members, which were clustered into three main groups. In total, 53 members were clustered in group A, 14 members in group B, and 6 members in group C (Figure 1). Noticeably, both CtF3′5′H1 and CtF3′5′H7 were clustered together to the largest group A, suggesting a close relationship with most of the members of other plant species. Importantly, all members of AtF3′5′H were clustered together in group C with no other members. The classification of CtF3′5′H1 showed close proximity towards the F3′5′H members of *Cynara cardunculus* and *Lactuca sativa*. Similarly, CtF3′5′H7 was closely located to F3′5′H members from *Cynara cardunculus* and *Artemisia annua* plant species, indicating the divergence of safflower *F3′5′H* genes at this point. On the other hand, group B including members of *Vitis vinifera*, *Actinidia chinensis*, *Camellia sinensis*, *Quercus suber*, *Ipomoea triloba*, and *Nicotiana tomentosiformis*. Conclusively, the phylogenetic analysis of putative safflower CtF3′5′H sequences demonstrated significant evolutionary relationship with previously identified *F3′5′H* in other plants. It indicated that safflower CtF3′5′Hs might coexist in the same functional domain with already-discovered F3′5′H from other plant species, largely ascribed to secondary metabolite biosynthesis, plant growth, and developmental processes.

### 2.2. Gene Structure, Protein Motif, and Cis-Regulatory Units

The gene structure organization of *CtF3′5′H* genes was investigated through Gene Structure Display Server 2.0 program. The results showed that gene structure of the *CtF3′5′H*1 and *CtF3′5′H*7 are relatively consistent with each other. However, the structure of *CtF3′5′H* was found to be more conserved than the *CtF3′5′H*7 because it consists of one intron and three exons whereas *CtF3′5′H*7 consists of one intron, three exons, and an additional UTR region. The gene structure organization of *CtF3′5′H* genes suggested positive implications in controlling the expression of their corresponding partner genes (Figure 2a). Similarly, the protein sequences of safflower *F3′5′H-*encoding genes were examined for their conserved protein motifs. The protein sequence of CtF3′5′H1 showed the presence of 10 highly conserved protein motifs. Noticeably, the motifs distribution of CtF3′5′H7 showed the absence of Motif4 (marked with purple color) as described in CtF3′5′H1. These motifs were almost similar in length with a range of 23 to 50 amino acids. In addition, through the position of these motifs, it was found that most of these motifs were located at both C-terminal and N-terminal. The cis-elements in the promoter region of two safflower *CtF3′5′H* genes showed several conserved cis-regulatory units that are known to be involved in photoreactions and light responses (G-Box elements), auxin-responsive elements (TGA-Box), low-temperature (LTR element), and abscisic acid (ABRE elements) (Figure 2c). The presence of these widely known cis-elements within the promoter region of *CtF3′5′H* genes highlighted their potential role in various biological pathways associated with plant growth and development, hormonal regulation and stress responses.

### 2.3. Tissue Specific Expression of Two CtF3′5′H Genes Revealed Differential Pattern of Transcriptional Regulation in Safflower

To confirm the expression level of *F3′5′H1* and *F3′5′H7* genes in different flowering stages of safflower (bud, initial, full, and fading), qRT-PCR analysis was conducted. Simultaneously, the accumulation pattern of total flavonoid content in these flowering stages was also investigated. The results showed that the expression of *F3′5′H1* was significantly increased at the first three flowering stages; however, the expression level was decreased during the decline flowering period (Figure 3). Importantly, the highest expression of *CtF3′5′H1* gene was detected at the full-blooming flowering stage, consistent with the total flavonoids content (Appendix A). Noticeably, the expression pattern of *CtF3′5′H7* showed an opposite trend suggesting down-regulation in the first three flowering stages and a significant increase at the full-bloom flowering stage. These findings revealed the differential expression and regulation of these genes through different stages of flower development in safflower. From these findings, it is also inferred that the differential regulation of *CtF3′5′H* genes through different flowering stages could also be associated with floral pigmentation ranging from yellowish to red in safflower.

### 2.4. Molecular Cloning and Subcellular Localization

The full-length *CtF3′5′H1* was successfully transformed into pEASY-T1 vector through heat and shock method, and the transformed colonies were confirmed with half-colony PCR and double restriction digestion system (Appendix A). The Sanger sequencing results verified that the amplified PCR product was compatible to the actual size of *CtF3′5′H1* without a single base mutation. The sequence analysis showed that CtF3′5′H1 encodes a 69.35 KDa polypeptide with a possible PI (Isoelectric point) of 5.01. Other physicochemical properties of CtF3′5′H1 protein showed 0.707, 28.63, and 33.27 for Grand average hydropathicity (GRAVY) values, aliphatic index, and instability index, respectively. The overall physical and chemical properties have demonstrated the stability and effectiveness of CtF3′5′H1 putative protein. The online software (http://www.csbio.sjtu.edu.cn/bioinf/plant-multi/) (accessed on 20 March 2021). was primarily used to predict the theoretical subcellular localization of the candidate CtF3′5′H1. The software-based localization signals were abundantly detected in endoplasmic reticulum followed by cell membrane and then nucleus. A deeper understanding of CtF3′5′H1 subcellular localization was further investigated by using a simple in transient expression system in *Nicotiana benthamiana.* The transient transformation of the young leaves of tobacco were transformed (via infiltration) with the recombinant (pCAMBIA1302-CtF3′5′H1-GFP) Agrobacterium *EHA105* strain containing a fused (Green fluorescent protein) *GFP* gene. The fluorescence of GFP signals was noticeably detected in the cell membrane of the infected tobacco cells during the observation under confocal laser scanning microscopy, suggesting that the translational construct of *CtF3′5′H1* could be identified as a membrane-bound protein (Figure 4). On the contrary, the result of the fluorescence emitted by the empty vector (control experiment) in tobacco cells indicated a diffused and scattered pattern of green signals throughout the cell. 

### 2.5. Generation of CtF3′5′H1 Overexpressed Arabidopsis Lines and Expression Analysis

In order to examine phenotypic and morphological changes between transgenic lines and wild-type plants, we chose two *CtF3′5′H1-*overexpressed transgenic lines (OE9 and OE10). Further analysis revealed that there was no discernible change in leaf size between the two transgenic lines (Figure 5a). However, we observed that the morphology of OE9 and OE10 were more elongated and curlier in shape, and phenotype is darker green in color than the wild type. The relative expression levels of ten overexpressed lines were significantly different, among which the expression level of OE9 and OE10 was relatively high. Furthermore, the content of anthocyanin and proanthocyanidins in the selected transgenic lines revealed that the anthocyanin content of OE9 and OE10 lines was higher reaching to it maximum (0.8 mg/g) than that of wild-type and the remaining transgenic lines. Noticeably, the proanthocyanidins content of the two lines was significantly lower than that of other lines (Figure 5b). The results were further verified by quantifying the total flavonoid content in transgenic lines with 500 ng/mL rutin as standard. Importantly, the total flavonoid content of transgenic lines OE9 and OE10 reached to its maximum 10.45 mg/g and 7.39 mg/g, respectively. Based on these findings, we inferred that *CtF3′5′H1* could be a key gene involved in the regulation of flavonoids biosynthesis in transgenic plants (Figure 5c).

### 2.6. Relative Expression of CtF3′5′H1 and Determination of Metabolites in Transgenic Arabidopsis under Various Abiotic Stresses

In order to determine the relative metabolite contents, the wild type, OE9 and OE10 transgenic lines were supplied with 50 mmol/L MeJA solution under variable time points. The results showed significant variations in leaf size, shape, and distinct phenotype between the two transgenic lines and wild-type Arabidopsis when MeJA treatment time was increased (Figure 6a). Furthermore, the expression level of *CtF3′5′H1* gene increased gradually with the increase in MeJA treatment time. The highest expression was detected in OE9 line followed by OE10 line compared with the wild-type. Similarly, the anthocyanin content in the selected transgenic lines showed an upward trend at different time points and reached to its maximum (0.65 mg/g) at 60 h in OE9 line, whereas the proanthocyanidins content showed an upward trend in transgenic lines compared with wild-type (Figure 6b). Moreover, the results demonstrated that the content of total flavonoid in OE9 and OE10 was relatively higher than the wild-type plant reaching to it maximum (12.87 mg/g) at 60 h time point in OE9. The results of total flavonoid content were further confirmed with HPLC analysis using rutin as a standard, suggesting the detection of increased flavonoid content in transgenic lines than the wild-type plants (Figure 6c).

Furthermore, the wild-type and transgenic Arabidopsis lines OE9 and OE10 were sprayed with a 100-mmol/L methyl violet solution with a time frame of every 12 h. The results showed significant phenotypic differences among the transgenic lines and wild-type Arabidopsis when the duration of MV stress was increased. From these findings, we implied that the reactive oxygen species resulting in the antioxidant activity in vivo was increased more than the wild-type as the transgenic plants demonstrated relatively greenish leaves phenotype, and exhibited lesser wilting state when the MV treatment time increased steadily (Figure 7a). In the next level, the expression pattern of *CtF3′5′H1* also showed up-regulation with the increase of treatment time and reached to its maximum at 48-h time points in both OE9 and OE10 transgenic lines than the wild-type under MV stress. The expression of *CtF3′5′H1* was down-regulated at 60-h time points, suggesting that long stress encounters may suppress the antioxidant potential of transgenic plants (Figure 7b). The content of anthocyanin in transgenic lines also demonstrated significant increase at 48 h time points and reached to its maximum (1.2 mg/g), whereas the proanthocyanidins content showed a downward trend in transgenic lines compared with wild type (Figure 7b). In addition, the total flavonoid content quantification resulted in an increased level reaching to its peak (2.76 mg/g) at 48 h in transgenic line OE9 followed by OE10 than the wild-type. The results of total flavonoid content were further confirmed with HPLC analysis using rutin as a standard, suggesting the detection of more increased flavonoid content in transgenic lines than the wild type plants (Figure 7c).

The seedlings of *A. thaliana* (WT, OE9, and OE10) on MS solid medium under UV-B light (284 nm) were irradiated for a period of 5 and 10 days, respectively. When exposed to UV-B light for five and ten days continuously, no significant phenotypic differences were observed in the leaves of transgenic plants, and the leaves were fully wilted and died on the 10th day of UV-B treatment (Figure 8a). However, the expression level of *CtF3′5′H1* demonstrated a gradual increase in OE9 and OE10 transgenic seedlings on the fifth day after UV-B treatment. On the contrary, the *CtF3′5′H1* expression was significantly down-regulated after 10 days of UV treatment in both transgenic seedlings compared to WT seedlings (Figure 8b). Similarly, the content of anthocyanin in OE9 and OE10 increased up to 0.83 mg/g after 5 days of UV-B treatment and then decreased to 0.69 mg/g after 10 days of UV-B treatment. Similarly, the content of proanthocyanidins continued to decrease and approached 0.1 mg/g on day 10 (Figure 8b). After 5 days of UV-B light stress, the content of total flavonoids increased up to 4.76 mg/g (Figure 8c), and then decreased after 10 days in both transgenic lines compared with wild-type plants. Altogether, these findings provide important insights into the identification of the putative role of *CtF3′5′H1* as a key regulator of abiotic stress tolerance by promoting anthocyanin accumulation in safflower.

### 2.7. Recombinant CtF3′5′H1 Protein Possess In Vitro Catalytic Activity

According to the predicted protein molecular weight, the size of a CtF3′5′H1 protein was calculated as approximately 69.35 kDa. The crude proteins extract was separated on 10% SDS-PAGE, and the target protein with an expected band size of (69 kDa with two HIS-TAG) was detected and visualized with Coomassie bright blue (Appendix A). The results of western blot detection showed that the target protein was expressed successfully (Appendix A). After purification, the eukaryotic expressed protein was used to conduct the in vitro enzymatic activity against various substrates of flavonoids. The HPLC-MS analysis revealed that CtF3′5′H1 was able to directly act on naringenin to generate eriodictyol with a conversion rate as high as 100% as compared to other substrates. The fragment ion *m*/*z*: 152.9 was used for naringin, and *m*/*z*: 289.1, 290 was detected for phenol. The highest activity of CtF3′5′H1 was shown at pH 5.0 (0.1% formic acid buffer) and 30 °C. Enzyme kinetic analysis showed that the km value of CtF3′5′H1 was 144.7 mM. These results indicated that CtF3′5′H1 has a high affinity for different substrates of flavonoids. One of the variables in this experiment is substrate concentration, seven substrates with varying concentration gradients of 5, 10, 25, and 50 mM. The results revealed that when the temperature and enzyme concentration are both constant, the greater the substrate concentration, the lower the enzyme conversion product content (Figure 9). Therefore, the substrate concentration of 5 ng/mL is selected to detect the enzyme activity. As shown in (Appendix A), the R^2^ = 0.991 is analyzed, which shows that the experiment is feasible. The km value of the enzyme is 144.7 mol/L, which indicates that the affinity between the enzyme and the substrate is high, which is the substrate concentration when the maximum reaction rate is half. The enzymatic kinetics of F3′5′H1 in safflower accords with the Michaelis equation. In order to verify the specific catalytic function of the enzyme, we found in the highest-performance liquid chromatography that the standard sample with a peak of 25.564 min was detected near 273 nm. After the binding of the enzyme with the substrate, a new peak area was detected at 290 nm, and the peak time was 20.3 min. With the change of substrate concentration, the peak area of this position also changed.

### 2.8. Virus-Induced Gene Silencing (VIGS) Assay of CtF3′5′H1 in Safflower

The safflower leaves at the period before the bud stage, and the mature leaves under the sepals were selected for VIGS analysis. The *CtF3′5′H1* coding regions were silenced using a virus-induced gene silencing (VIGS) approach and then injected in the aforesaid leaf materials. In safflower leaves, the expression of TRV2-bait decreased significantly, which proved that the suppression was successful (Figure 10). Moreover, the content of proanthocyanidins and anthocyanins decreased dramatically. The phenotype of the leaves near sepals were yellowish green and curled compared to control plants transformed with the Tobacco Rattle Virus (TRV) empty vector. These results suggested that *CtF3′5′H1* might be involved in regulation of floral pigmentation. Further verification was performed by analyzing the changes of expression level of core structural genes before and after silencing at the middle and downstream of the pathway regulated by *CtF3′5′H1*. The downstream genes after silencing were down-regulated compared with those before silencing. The expression levels of *F3′H* gene (KP300884) after silencing decreased dramatically, while the expression levels of *dihydroflavonol-4-reductase* (*DFR*: OP616395) and *flavonol synthase* (*FLS*: KP300884) were relatively high compared with other genes (which are cloned in previous research) after silencing. Similarly, up to some extent, the proanthocyanidins in the tissue were reduced to half of the original after silencing, causing the leaves around the sepals to turn yellow. These findings importantly highlight the positive role of CtF3′5′H1 during the regulation of floral pigmentation in safflower.

### 2.9. Quantification of Total Flavonoids Content during VIGS by HPLC under MeJA Stress

The total metabolite content was determined in *CtF3′5′H1* suppressed leaves of safflower under normal and MeJA treatment. Under normal condition, the total flavonoid content of in suppressed leaves of safflower demonstrated a more significant decrease than those in non-silenced safflower leaves. The results were further confirmed with HPLC analysis suggesting that the total flavonoids of un-injected (non-silenced) safflower leaves were higher than those infected with different VIGS vectors (Figure 11a–c). In addition, the suppressed leaves of safflower were subjected to different concentrations of MeJA including 0, 5, and 10 mmol/L concentrations. The total flavonoid content elicited by 10 mmol/L MeJA concentration increased from 20.75 mg/g to 25.69 mg/g compared with 0 mmol/L treatment group (Figure 11d). These findings importantly highlighted the MeJA-induced regulation of plant-derived metabolites in safflower.

## 3. Discussion

In nature, flavonoids are found in the form of flavones, flavonols, proanthocyanidins, and anthocyanins, which are the most common types of flavonoids. In the face of a wide range of environmental and biological threats, these compounds serve vital functions in plant defense systems [32]. Anthocyanins are mainly responsible for their pink, red, blue, and purple colors and also assist plant systems to entice pollinators and cope with abiotic stress response [33]. Anthocyanin and its derivatives also present a large antioxidative capacity, allowing plants to combat ROS-induced risks. In comparison to anthocyanin, which is a well-known colored plant pigment in plants, flavones and flavonols are significant colorless polyphenolic compounds that are involved in plant defense against abiotic challenges, such as Ultraviolet-B (UV-B) stress [34]. Biological characteristics of flavonoids are dependent on the hydroxylation pattern of their B-rings. Antioxidant activity is connected to heightened anthocyanin B-ring hydroxylation, which turns them from pink or red to purple or blue [35]. At the flavanone and dihydroflavonol stages of biosynthesis, unique cytochrome P450 monooxygenases induce hydroxylation at the 3′ and 5′ sites. The aim of this study was to identify and characterize the role of a putative abiotic stress responsive *CtF3′5′H*1 gene from safflower, which promotes anthocyanin accumulation and regulation of other pigments in safflower.

The full-length *CtF3′5′H1* sequence was obtained from the safflower genome and subsequently cloned using gene-specific primers. The deduced amino acid sequence confirmed obvious homology with other previously identified F3′5′H from different plants. After efficient cloning, the subcellular localization of transiently expressed *CtF3′5′H1* in *N. benthamiana* leaves suggested a fluorescence pattern emerged in the cell membrane, confirming that CtF3′5′H1 is a membrane-bound protein. Previous investigations of flavonoid biosynthesis enzymes revealed that the vast majority of these enzymes were found in the cytosol and nucleus, with only a small number found in the vacuole and endoplasmic reticulum. Understanding the subcellular localization of key flavonoid biosynthetic enzymes can provide a deeper insight to their diverse pattern of transportation, transit, and subsequent trafficking to multiple cellular compartments. Similarly, the expression level of *CtF3′5′H1* was significantly up-regulated through different flowering stages including bud, initial, and full-blooming flowering, whereas its expression was down-regulated in the decline flowering period (Figure 4). These results suggested that the differential expression and regulation of *CtF3′5′H1* during different developmental stages of safflower is likely influenced by the regulation of color pigments from yellowish to red. Previous investigations also revealed that the expression of genes encoding *F3′5′Hs* have a significant impact on the accumulation level of flavonoids due to a wide range of B-ring hydroxylation patterns [16]. Furthermore, a wide range of MYB transcription factors were also demonstrated to control the expression of *F3′H* and *F3′5′H* encoding genes during anthocyanin biosynthesis and ripening of grapes [36]. These results importantly highlighted that the expression of F3′5′H encoding genes are tightly regulated with the accumulation level of the hydroxylated flavonoids during flower development in safflower.

The biosynthesis of anthocyanin is controlled on a fine scale both temporally and spatially, and it requires multiple channels of regulation, including transcriptional and post-translational mechanisms [34]. Phytohormones, in particular, play a critical role in controlling plant flavonoid biosynthesis. Many studies have investigated the effectiveness of plant hormones on flavonoid accumulation including cytokinin [37], abscisic acid [38], ethylene [39], and jasmonate [40]. Similarly, in most of the fruit species, such as strawberry [41], grape [42], and blueberry [43], the MeJA induction demonstrated increased flavonoid accumulation. In this study, we also investigated MeJA-induced regulation of anthocyanin biosynthesis in *CtF3′5′H1* overexpressed transgenic lines. It was shown that anthocyanin levels increased with different treatment time, peaking at 0.65 mg/g in the OE-9 line after 60 h, while proanthocyanidins levels decreased in the transgenic lines relative to the wild type. Noticeably, the expression level of *CtF3′5′H1* was simultaneously up-regulated in the OE-9 transgenic line with the increase in MeJA treatment time compared to wild-type. From these findings, we implied that MeJA induces the accumulation of anthocyanin through *CtF3′5′H1*-regulated expression. In addition, the content of anthocyanin in transgenic lines under methyl violet stress and UV-B irritation was also demonstrated to be significantly increased in OE lines, whereas the proanthocyanidins content showed a downward trend compared with wild type. The UV-B-absorbing flavonoids are an efficient non-enzymatic strategy to attenuate photoinhibitory and photooxidative damage produced by UV-B stress by either limiting UV-B radiation penetration or quenching reactive oxygen species (ROS). Altogether, these findings provide important insights into MeJA-induced anthocyanin accumulation most probably mediated by the enhanced expression of *CtF3′5′H1* gene that controls a key hydroxylation step in the anthocyanin biosynthetic pathway.

Flavonoid biosynthetic pathway enzymes have varied substrate selectivity, which has been shown to have a major impact on the composition of flavonoid compounds in different plant species. In this study, *CtF3′5′H1* was heterologously expressed using the yeast expression system (pYES2/CT), and then the in vitro activity was carried out with different substrates using HPLC-MS analysis. The results showed that *CtF3′5′H1* was able to directly act on naringenin in order to produce eriodictyol with a conversion rate as high as 100% as compared to other substrates. The rate at which an F3′5′H enzyme performs hydroxylation depends on the reductase utilized in the expression system. In petunias, cytochrome b5 has also been shown to be necessary for the complete action of F3′5′H [44]. Similarly, a targeted transposon mutation inactivated the *cytochrome-b5* gene, resulting in the decreased F3′5′H activity and a decrease in the accumulation of anthocyanin with a 5′ substitution, altering floral coloration. Our results were found consistent with the previous findings of [19], who also suggested that F3′5′H in petunia had highest enzymatic activity when using naringenin as a substrate. These findings were further verified by conducting the Virus-induced gene-silencing (VIGS) assay of *CtF3′5′H1* by suppressing a 300 bp fragment in the coding region of *CtF3′5′H1* gene in safflower. As a result, anthocyanin and proanthocyanidins levels were considerably reduced in silenced tissues, and the expression of TRV2-bait dropped significantly. Moreover, the phenotype of the leaves near sepals were yellowish green and curled. Noticeably, the downstream genes after silencing were down-regulated compared with those before silencing. For instance, the expression level of the *F3′H* gene after silencing decreased dramatically, while the expression level of DFR and FLS were relatively high compared with other genes after silencing. Similarly, up to some extent, the level of proanthocyanidins in the leaves’ tissue were reduced to half of the original after silencing, causing the leaves around the sepals to turn yellow. Furthermore, the metabolite accumulation was determined in *CtF3′5′H*1 suppressed leaves of safflower under normal and MeJA treatment. The results indicated that anthocyanin content in suppressed leaves demonstrated a significant decrease than those in wild-type safflower without MeJA treatment. However, when the leaves of safflower were subjected to different concentrations of MeJA treatment, the accumulation of total flavonoid content was increased from 20.75 mg/g to 25.69 mg/g compared with 0 mmol/L treatment group. These findings importantly highlight the positive role of MeJA enhanced expression of *CtF3′5′H*1 during the regulation of floral pigments in safflower and provide a basis for designing efficient biocatalysts for achieving highly efficient and directed biosynthesis of important bioactive flavonoid glycosides.

## 4. Materials and Methods

### 4.1. Plant Materials, Vectors and Strains

In this study, “Jihong No. 1” safflower was planted and sampled in our laboratory’s artificial chamber under controlled conditions (25 °C, 16 h light and 8 h dark) until the flowering stage in the bioreactor and drug development engineering research center base of the Ministry of Education of Jilin Agricultural University. The plant material is annual with about 20 g. The flower petals of bud stage, initial stage, full-bloom stage, and fading stage were collected, immediately put into a transparent bag, marked and wrapped with tin paper, put into liquid nitrogen, and kept in −80 °C refrigerator until the next use. The carriers pCAMBIA1302-GFP and pYES2-NTC were previously stored in the laboratory.

### 4.2. Genome-Wide Identification of CtF3′5′H Genes from Safflower

A complete F3′5′H HMM domain was retrieved from the Pfam database online server, available at (http://pfam.xfam.org/) accessed on 15 February 2021 [45]. Following the HMM Profile of CtF3′5′H protein, the scanning of HMMER (http:/hmmer.janelia.org/) accessed on 15 February 2021 is used against Ji Hong NO. 1 safflower (https:/www.ncbi.nlm.gov/bioproject/PRJNA399628) accessed on 15 February 2021 with an E-value of less than 0.01 using a strict condition. Sequences other than safflower F3′5′H proteins were extracted from (NCBI https://www.ncbi.nlm.nih.gov/) accessed on 15 February 2021 and the Arabidopsis Information Resource (TAIR) (http://www.Arabidopsis.org/) accessed on 15 February 2021.Furthermore, the physicochemical properties of safflower F3′5′H proteins were investigated with the help of different online tools. The online webserver of ExPASy Prot Param online tool (https://web.expasy.org/protparam/) accessed on 15 February 2021 was used to calculate the theoretical isoelectric point (PI), protein size, grand average of hydropathy (GRAVY), and theoretical molecular weight (MW). The subcellular localization of each gene was predicted with the help of the Cello web server (http://cello.life.nctu.edu.tw/) accessed on 15 February 2021 and WoLF PSORT webtool.

### 4.3. Phylogenetic Analysis

The full-length protein sequences of safflower F3′5′H and from different plant species including *A. thaliana*, *Artemisia annua*, *Pyrethryum cinerariifolium*, *Lactuca saligna*, *Vitis vinifera*, *Mikania micrantha*, *Helianthus annuus*, *Camellia sinensis*, *Cynara cardunculus var. scolymus*, *Lactuca sativa*, *Nicotiana tomentosiformis*, *Ipomoea trilobal*, *Quercus suber*, and *Ipomoea nil* were extracted from NCBI and TAIR Database. The sequences were aligned with Clustal W, and a phylogenetic tree was generated using MEGA-11 software following the Neighbor-Joining (NJ) method with 1000 bootstrap replicates. The tree was visualized with the ITOL program (Wang et al., 2021a). The subgroups are marked by different colored lines in the tree.

### 4.4. Gene Structure, Conserved Motifs, and Promoter Analysis

The CDs and genomic sequences of *CtF3′5′H* genes were analyzed by the online webserver of GSDS (Gene Structure Display Server (http://gsds.cbi.pku.edu.cn/index.php) accessed on 15 February 2021 to observe the organization of gene structures. The online MEME web server (4.8.1) accessible on http:/meme.nbcr.net/mème/cgi-bin/meme.cgi) accessed on 15 February 2021 was used for the identification and screening, and positioning of the conserved protein motifs of CtF3′5′H proteins. The optimization parameters were as follows: 0 or 1, each sequence has a base sequence; 10 bp, the base sequence width range; and the other three large-scale base sequences. All different parameters use default values. The graphical representation of the widely spread protein motifs was created with the help of EvolView v.2 software. Furthermore, the 2 kb upstream 5` UTR region of each selected *CtF3′5′H* gene was used for investigating the distribution of cis-elements within the promoter region using PLACE available at (https://sogo.dna.affrc.go.jp/) accessed on 15 February 2021.

### 4.5. Expression Analysis of CtF3′5′H1 and CtF3′5′H7 in Safflower

The flower petals at different flowering stages of safflower grown in our laboratory’s artificial chamber under controlled conditions (25 °C, 16 h light and 8 h dark) were collected. The total RNA content was simultaneously isolated from the flower tissues at (bud stage, initial flowering stage, full-bloom stage, and decline stage). The quantitative real-time PCR assay was carried out to examine the expression level of *CtF3′5′H1* and *CtF3′5′H*7 genes. The expression level of each transcript was normalized with 18 s ribosomal RNA gene. The measurement of the relative expression level was calculated following the 2^−ΔΔCT^ method. Three biological repeats for each gene were used. The primer details are listed in Appendix A.

### 4.6. Molecular Cloning of CtF3′5′H1 and GFP Translational Fusion Construction

The full-length cDNA sequence of *CtF3′5′H*1 gene was amplified using the gene specific primers CtF3′5′H1-R: ATGATACAAAACAGCATCTGG (5′-3′) and CtF3′5′H1-F: TCATAAGTAGAGGTTTGTGT (5′-3′). The genomic sequence of *CtF3′5′H*1 is available under the accession number (MW436618) in the public database of NCBI. The full-length sequence of *CtF3′5′H*1 (1596 bp) gene was amplified through polymerase chain reaction using Pfu DNA Polymerase (Takara, Beijing, China). After the cloning, the desired fragment was transformed into the pEASY-T1 vector (Takara, Dalian, China) for sequencing. After sequence confirmation, the vector (pCAMBIA1302-GFP) under the control of CaMV35S promoter was used to examine the experimental validation of *CtF3′5′H*1 subcellular localization using *N. benthamiana* fresh apical leaves (He et al., 2019). The recombinant vector was further introduced into the Agrobacterium strain *EHA105* using the direct transformation method. At the same time, the empty carrier of (pCAMBIA1302-GFP) alone was also transformed into the *EHA105* strain. A single colony of Agrobacterium strain (EHA105) expressing *CtF3′5′H*1 along with an empty vector-containing colony was grown at an OD_600_ of 0.8 (optimal), and then collected through centrifugation. The resuspension of bacterial culture was subjected to a solution containing acetosyringone (0.1 M), MgCl_2_ (0.5 M), and MES (0.5 M), with pH adjusted at the number of 5.6 (Wang et al., 2021b). The healthy anterior leaves of tobacco were pierced with a needle to maximize the penetration of Agrobacterium infection. The treated leaves of both strains were placed under a laser scanning confocal microscope (Leica TCS SP5) after 36 h of transformation. 

### 4.7. Generation of Transgenic Arabidopsis via Floral Dip Transformation

To acquire the stable genetic transformation of *CtF3′5′H1* in Arabidopsis, the plant overexpression vector pCAMBIA3301 was manipulated following digestion with *Nco* I and *Not* I restriction enzymes, and then the full-length sequence of *CtF3′5′H*1 treated with aforesaid enzymes was ligated into the pCAMBIA3301 vector. The recombinant vector of pCAMBIA3301 harboring *CtF3′5′H1* was transformed into Agrobacterium tumefaciens EHA105 strain using the heat shock method, and then transformation of wild-type Arabidopsis was conducted following the floral dip infiltration method. For molecular and phenotypic investigations, we used T3 Arabidopsis lines and (WT) Arabidopsis (control). According to the weight of the sample (1 g sample is added with 3.5 mL extract, which can be added according to different materials), we prepared the extract on ice, ground the sample in a mortar with liquid nitrogen, added the extract, and allowed to rest on the ice (3–4 h). The supernatant was extracted by centrifugation at 8000 rpm for 40 min at 4 °C or 11,000 rpm for 20 min at 4 °C, and the sample was prepared. Protein extract: 300 mL, 1 M tris-HCl (pH8), 45 mL, Glycerol 75 mL, and Polyvinylpolypyrrordone 6 g.

### 4.8. Abiotic Stress Elicitation of CtF3′5′H1-Overexpressed Transgenic Arabidopsis

Methyl jasmonate (Beijing Solarbio Science & Technology Co. Ltd., Beijing, China) solution with a concentration of (50 mmol/L) was prepared, and a total of 100 mL solution was applied to wild-type and two overexpressed transgenic lines, respectively. The time period for spraying methyl jasmonate was set according to a 12-h cycle and increased up to 60 h, consecutively. In the case of methyl violet stress, a final concentration of 50 mmol/L methyl violet (Beijing Solarbio Science & Technology Co.,Ltd.) solution containing Tween-20 and reverse osmosis water was prepared. The solution was completely protected from light at low temperature in order to ensure its biological activity. The time duration from 12 h was gradually raised to 60 h. Both wild-type and two overexpressed transgenic lines were sprayed three times for each treatment. For UV-B stress, the seedlings of *A. thaliana* (WT, OE9, and OE10) grown on MS solid medium were chosen. To provide continuous lighting, the petri dish was opened and treated with UV-B LED Light-Emitting Diode Clear Round Lens with intensity (UV-B, 80 W/cm^2^,100 µW/cm^2^) and a wavelength of 284 nm. After the UV-B illumination, the growth of Arabidopsis seedlings on day 5 and 10 was observed. Finally, total RNA extraction, reverse transcription, and quantitative real-time polymerase chain reaction (qRT-PCR) analysis were conducted from each experimental group, respectively. Furthermore, the quantification and extraction of total flavonoid content, anthocyanin, and proanthocyanidins was also carried out in all treated groups. 

### 4.9. Quantitative Analysis of Anthocyanin, Proanthocyanidins and Total Flavonoid Content

A total of 0.25 g fresh plant samples was used for anthocyanin extraction and quantification. For this purpose, the extraction solution containing (50 µL hydrochloric acid + 4.5 mL methanol) was added and well-mixed after the plant material was fully converted to a fine powder in liquid nitrogen. Plant samples were then placed in darkness at 4 °C for 16 h before being centrifuged at 4 °C for 10 min at 10,000 rpm at 4 °C. Following the addition of 100 L of the supernatant to the 96-well plate, the absorbance values at 530 nm and 657 nm were determined and subsequently replaced into the formula 2 (A_530_ nm-0.25A_657_ nm). The solution without the plant extract served as the control, and all of the results were determined using three different biological replicates. Similarly, the quantification of proanthocyanidins was carried out using 0.1 g fresh weight plant materials. A total volume of 2 mL extract solution containing (acetone/water/glacial acetic acid = 150:49:1) was added to the plant material and were allowed to be ground into a slurry solution, followed by continuous shaking for 1 h at 28 °C and then centrifugation for 10 min at 10,000 rpm at room temperature. We took 500 µL of supernatant from the previous extract and added 500 µL of anhydrous ethanol with an 80% concentration. We homogenized it completely, and then separated 500 µL from the previous mixture and added 1.5 mL of dimethyl formamide. The quantification of proanthocyanidins was calculated according to the formula 7 (241.19OD-6.1902)/100 at an absorption peak of 640 nm. Distilled water was used as a blank control, and three biological replicates for each treatment group were used.

Similarly, the total flavonoid content was determined using a fresh weight 0.1 g each plant material. The plant tissues were ground in liquid nitrogen and then transferred to the test tubes followed by adding 500 µL methanol, 100 µL 8% sodium nitrite solution, and dissolved completely for 5 min. After that, we added 100 µL of aluminum nitrate solution (15%) and let stand for a period of 5 min, then added 1 mL sodium hydroxide solution (6%), followed by adding 6 mL distilled water, and the mixture was fully mixed by vortex oscillation and left for 3 min. Finally, the supernatant was transferred into a 2-mL centrifuge tube and centrifuged at 13,000 rpm for 10 min. The supernatant was immersed in an ultrasonic cleaning equipment at 50 °C for 30 min. Then, the solution was filtered using a 0.22 µm filter, and the quantification of total flavonoid content was observed using the absorbance spectra ranging from 300–500 nm using UV-Vis spectrophotometer.

### 4.10. Recombinant Protein Expression and Purification

The PYES2-NTC expression vector under the control of GAL 1 promoter was used to identify the expression of *CtF3′5′H*1 using INVSc1 competent cells (Yamamoto et al., 2021). The INVSc1 competent cells were melted on the ice and then added to the pre-cooled plasmid containing 2–5 μg Magi Carrier DNA (95–100 °C, 5 min, rapid ice bath, repeated once). Then, a total of 10 μL PEG-LiAc500 μL solution was prepared and mixed several times, in a water bath at 30 °C for 30 min (flipped 6–8 times at 15 min) and then at 42 °C for 15 min (turn 6–8 times at 7.5 min). The supernatant was discarded by 5000 rpm centrifugation for 40 s, resuspended with 400 μL ddH_2_O, and discarded after 30 s centrifugation. Using ddH_2_O 50 μL heavy suspension, plasmid screening was performed on SD-URA medium plate cultured at 29 °C for 48–96 h. The transformants selected from SD-URA solid medium were inoculated in SD-URA liquid medium at 30 °C for 48 h. The bacterial liquid cultured in 2 mL was centrifuged by 13,000 rpm for 1 min, then the supernatant was re-suspended in 10 μL sterile water, boiled in hot water for 10 min, placed in liquid nitrogen for 5 min, then boiled in hot water for 5 min and centrifuged at 13,000 rpm for 1 min. Galactose was used to induce the expression of the target protein. SDS-PAGE and western blotting were used to detect whether the expression of the protein was correct and whether it was successfully transferred into yeast system.

### 4.11. In Vitro Enzyme Activity Assays of Recombinant CtF3′5′H1 Proteins

The correctly expressed protein was purified by using beaver biological His-tag purification reagent. The reaction mixture containing (5 ng/mL naringenin, recombinant F3′5′H1 protein) was analyzed by LC/MS analysis at 30 °C for 25 min. Heat-inactivated enzymes (100 °C for 15 min) were used as the negative control. The change of the content of the product was detected every 5 min, and the optimum reaction condition of enzyme activity was reached by 25 min, and the pH was 5.0 under this condition. The standard curves of different peak areas at different concentrations were prepared by external standard method, and the samples were treated and detected in parallel with the experimental groups of unknown products and enzymes. 

### 4.12. Suppression of Agrobacterium Tumefaciens Infection of Safflower

The *Agrobacterium tumefaciens* strain GV3101 was used for VIGS analysis. Electrocompetent *A. tumefaciens* was transformed with pTRV1 and pTRV2 vectors following the instructions of [46]. Transformants were screened by PCR, which contained the expected construct of pTRV1 and pTRV2 vectors. Positive transformants were used to inoculate on 100 mL YEP containing 100 μg mL^−1^ kanamycin, 50 μg mL^−1^ rifampicin, which were grown for 40 h at 28 °C. These cultures were pelleted at 5000 rpm for 10 min and resuspended in infiltration solution containing 15 mM MES and 10 mM MgCl_2_. The A. tumefaciens strains harboring pTRV2 constructs were then mixed in a 1:1 ratio with strains pTRV1, and this mixture was used to inoculate the plants. Two-month-old plants of safflower were inoculated with the A. tumefaciens culture by a syringe by introducing a small wound with a needle, but not disturbing the leaf morphology. Each experimental group should be injected with 20 leaves. After 15 days of infection, the leaves were collected for analysis.

### 4.13. Safflower Leaves under Methyl Jasmonate Stress after Gene Silencing

Approximately 100 mM methyl jasmonate (MeJA) solution was used to stress the safflower leaves harboring Col-0 vector, pTRV1-pTRV2 vectors, pTRV1-pTRV2-bait vector, simultaneously for 30 min. The treatment was repeated thrice by avoiding light encounter for 24 h. After retaining the normal growth, the leaves’ tissues were obtained under methyl jasmonate stress, and total flavonoids content was quantified following the above-mentioned protocol as well as (High Performance Liquid Chromatography) HPLC analysis.

### 4.14. Statistical Analysis

The one-way analysis of variance (ANOVA) was used to calculate the statistically significant differences between the means of three different independent biological replicates. The package of Statistix 8.1 software was exploited. The *p*-values equal to 0.05 were kept statistically significant. The relative expression level was calculated following the 2^−ΔΔCT^ method. Calculation formula of anthocyanin content: 2(A530-0.25A657), Calculation formula of proanthocyanidins content: 7(241.19OD-6.1902)/100, unit: mg/g. The content of total flavonoids was the product ratio of peak area and concentration.

## 5. Conclusions

In this study, we presented the identification and molecular characterization of the candidate *CtF3′5′H1* gene likely involved in methyl jasmonate-induced regulation of anthocyanin accumulation in safflower. The expression analysis of *CtF3′5′H1* revealed a differential regulation pattern through different stages of flower development in safflower. Similarly, the overexpression of *CtF3′5′H*1 in Arabidopsis significantly enhanced anthocyanin accumulation under MeJA elicitation. The heterologously expressed *CtF3′5′H1* protein was able to catalyze different substrates of flavonoids, whereas the suppression of *CtF3′5′H1* in safflower via virus-induced gene silencing (VIGS) assay leads to decreased content of proanthocyanidins and anthocyanin as well as the down-regulation of key pathway genes involved in anthocyanin biosynthesis. Noticeably, MeJA-induced regulated of anthocyanins and flavonoids content in *CtF3′5′H1* silenced safflower was retained. Conclusively, these findings revealed important insights into the important role of *CtF3′5′H1* in MeJA-induced anthocyanin accumulation in safflower, providing new gateways towards the improvement of new safflower cultivars with improved metabolic properties in the future studies.

## Figures and Tables

**Figure 1 molecules-28-03205-f001:**
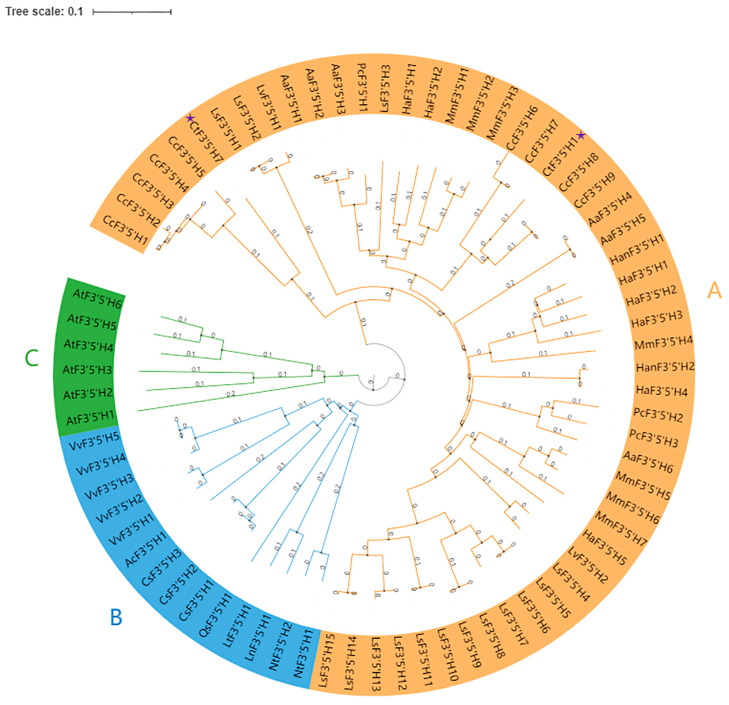
Phylogenetic analysis of F3’5’H sequences from safflower and other plant species. The scale bar represents 0.1 substitutions per site. The two candidate safflower genes are marked with purple stars. All members of F3’5’H were divided into three subgroups including (**A**) containing 53 F3’5’H members shown with orange color (**B**) containing 14 F3’5’H members shown with blue color and (**C**) containing 6 F3’5’H members shown with green color.

**Figure 2 molecules-28-03205-f002:**
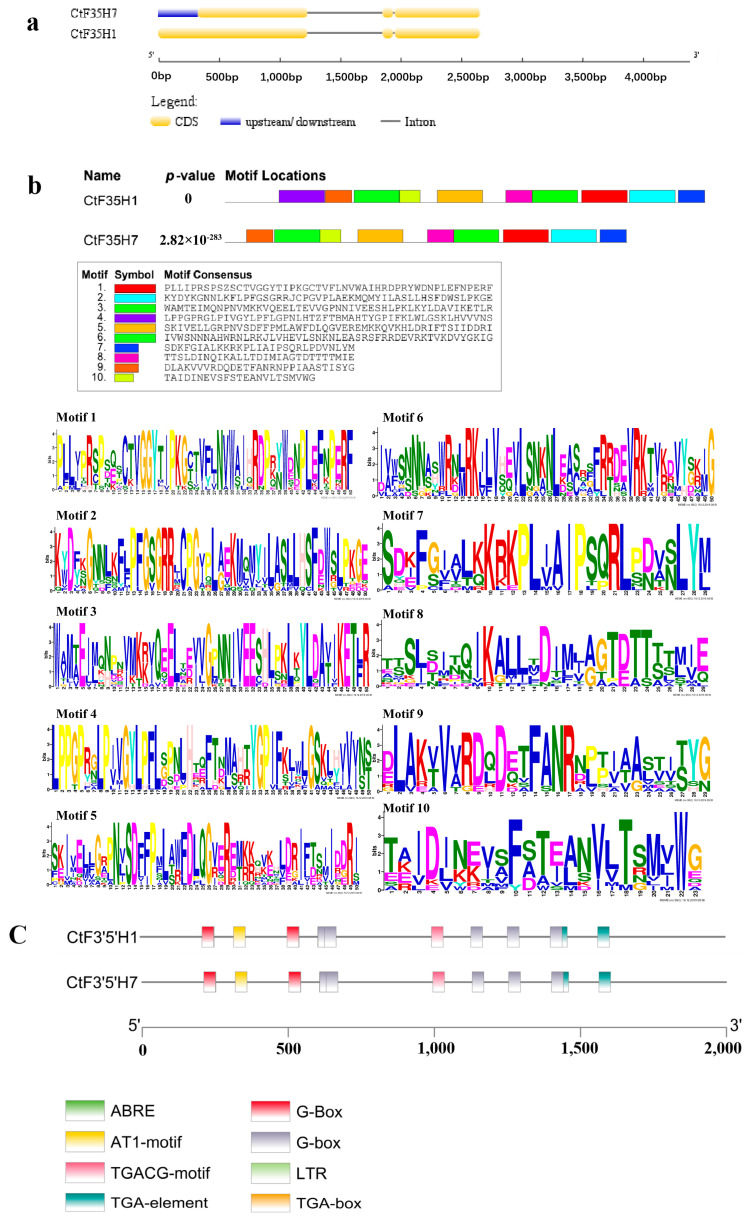
Analysis of gene structure, conserved motif, and domain of two candidate F3′5′H from safflower. (**a**) Gene structure organization. CDSs, upstream/downstream and introns are marked with yellow, blue and black boxes, respectively (**b**) The identification and distribution of conserved protein motifs of safflower F3′5′H using the online web server of MEME. A sum of 10 conserved motifs were obtained and demonstrated with different colors. The existence of each colored block at distinct position represents the position of the conservation and its matching motifs. (**c**) The schematic representation of eight cis-regulatory elements detected in the promoter regions of the two candidate *CtF3′5′H*-encoding genes in safflower.

**Figure 3 molecules-28-03205-f003:**
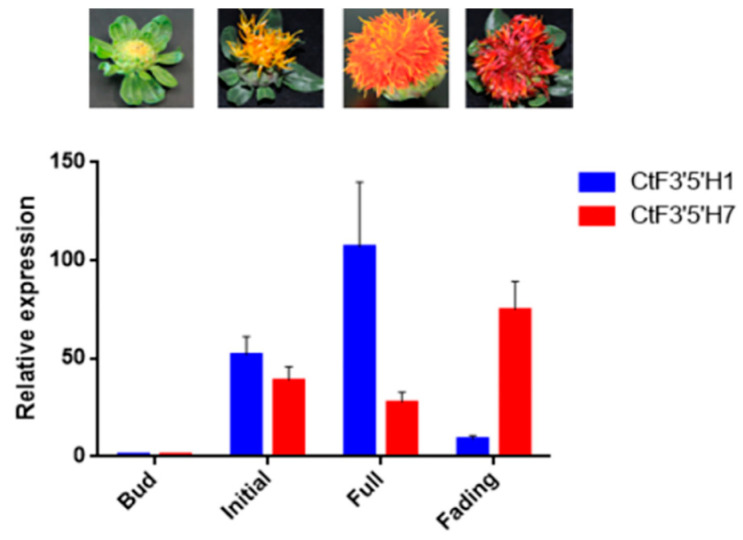
The qRT-PCR assay of two *CtF3′5′H* genes in four different flowering phases of safflower. The relative expression level was standardized with 18S rRNA gene (internal reference gene) and flower bud stage. The error bars show the differences of three independent biological replicates. Different colors represent different genes.

**Figure 4 molecules-28-03205-f004:**
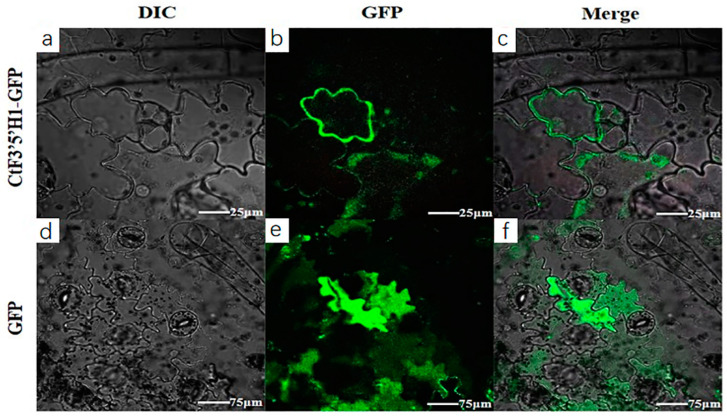
Subcellular localization of CtF3′5′H1 using transient expression of the GFP translational fusion (pCAMBIA1302-CtF3′5′H1-GFP) in tobacco apical leaves. (**a**–**c**) The fluorescence detection of (pCAMBIA1302-CtF3′5′H1-GFP) construct in the tobacco cells. (**d**–**f**) Transient expression of empty (pCAMBIA1302-GFP) vector. (**a**,**d**) Cell morphology in the bright field. (**b**,**e**) Demonstrate the dark field to visualize the GFP signals. (**c**,**f**) The images presented in overlaps field, respectively. The Leica confocal microscope (Leica TCS SP5) was used for fluorescence detection.

**Figure 5 molecules-28-03205-f005:**
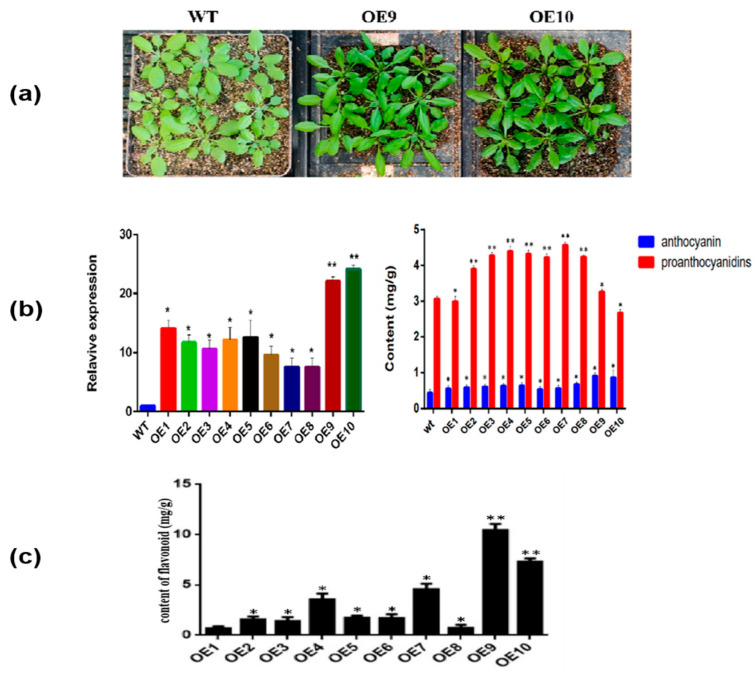
Characterization of *CtF3′5′H1* overexpressed transgenic Arabidopsis. (**a**) Phenotypic analysis of transgenic and WT plants. Using constant photographic conditions and light intensity, the transgenic plants demonstrated dark greenish leaves with elongated and curlier morphology than the WT. (**b**) Analysis of relative gene expression and anthocyanin proanthocyanin content in transgenic and WT plants. The relative expression level of *CtF3′5′H1* and accumulation of anthocyanin proanthocyanin content were relatively high in OE9 and OE10 lines compared to WT and remaining transgenic lines. (**c**) Quantification of the total flavonoid content in different transgenic lines. One-way ANOVA was used to calculate statistical significance among different groups. The asterisks represent (*: *p* > 0.05, **: 0.01 < *p* < 0.05).

**Figure 6 molecules-28-03205-f006:**
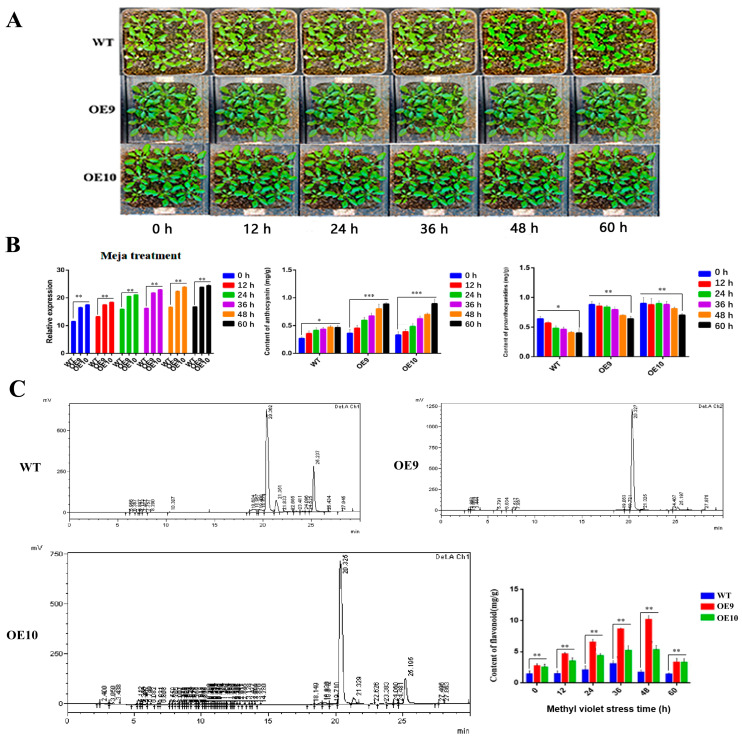
Phenotypic analysis, relative expression, and metabolites accumulation in *CtF3′5′H1* overexpressed transgenic lines under methyl jasmonate stress. (**A**) Phenotypic change under MeJA stress at different treatment times. MeJA treatment induced significant changes in leaf size, shape, and color in transgenic lines than WT plants. (**B**) *CtF3′5′H1* gene expression and content of anthocyanin and proanthocyanin analysis. MeJA treatment steadily enhanced *CtF3′5′H1* expression and anthocyanin and proanthocyanin accumulation with the increase in treatment time in both OE9 and OE10 lines than the WT plants. (**C**) The metabolite profile of OE9 and OE10 transgenic lines showed greater content of total flavonoid accumulation than the WT, peaking at 12.87 mg/g at 60 h. The HPLC analysis also confirmed that the transgenic lines showed higher accumulation of flavonoids than WT. One-way ANOVA was used to calculate statistical significance among different groups. The asterisks represent (*: *p* > 0.05, **: 0.01 < *p* < 0.05, ***: *p* < 0.01).

**Figure 7 molecules-28-03205-f007:**
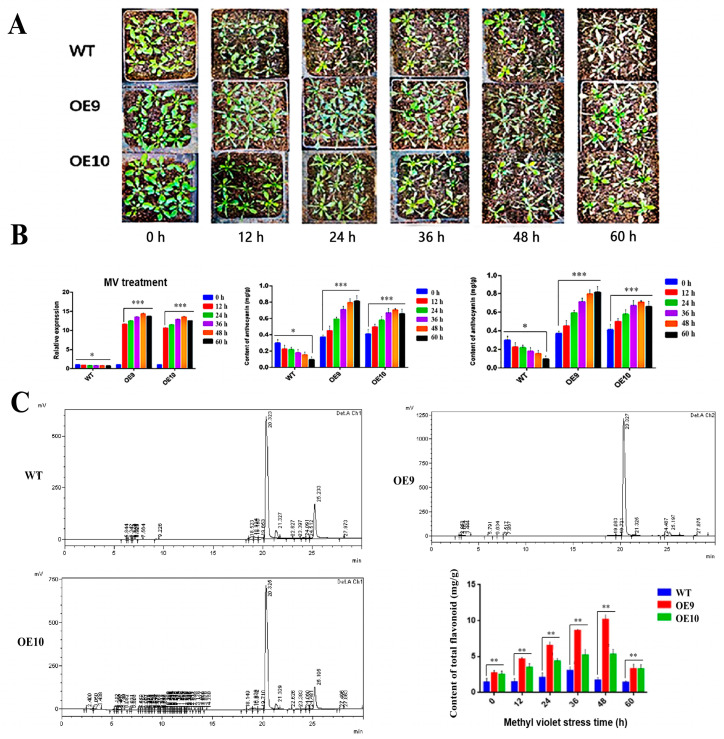
Phenotypic analysis, relative expression, and metabolites accumulation in *CtF3′5′H1* overexpressed transgenic lines under methyl violet (MV) stress. (**A**) Phenotypic changes under MV stress at different treatment times. Transgenic lines (OE9 and OE10) demonstrated relatively greenish leaves and exhibited lesser wilting state under MV induction at different time periods compared to WT plants. (**B**) *CtF3′5′H1* gene expression and content of anthocyanin and proanthocyanin analysis under MV stress. The expression of *CtF3′5′H1* indicated up-regulation with the increase of MV treatment time in OE9 and OE10 transgenic lines than WT plants. Anthocyanin and proanthocyanin quantification demonstrated more increased accumulation in transgenic plants than the WT plants. (**C**) The metabolite profile of OE9 and OE10 showed greater content of total flavonoid accumulation than the WT, peaking at (2.76 mg/g) at 48 h in OE9. The HPLC analysis also corroborated that transgenic lines showed higher flavonoid content than WT plants. One-way ANOVA was used to calculate statistical significance among different groups. The asterisks represent (*: *p* > 0.05, **: 0.01 < *p* < 0.05, ***: *p* < 0.01).

**Figure 8 molecules-28-03205-f008:**
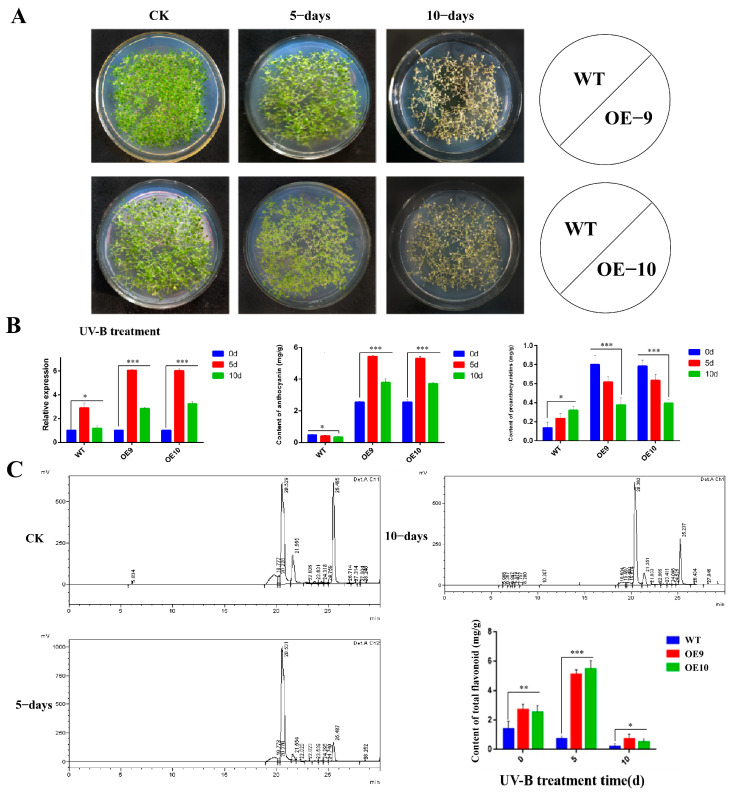
Phenotypic analysis, relative expression, and metabolites accumulation in *CtF3′5′H1* overexpressed transgenic lines under UV-B irradiation. (**A**) Phenotypic changes between two transgenic lines and WT seedlings on the 5th and 10th day after UV-B treatment on MS medium. Under UV-B light treatment, no significant phenotypic differences were observed in transgenic and WT seedlings. (**B**) *CtF3′5′H1* gene expression and content of anthocyanin and proanthocyanin analysis under UV-B irradiation. The *CtF3′5′H1* expression and content accumulation of both anthocyanin and proanthocyanin were increased after 5 days of UV-B treatment in transgenic seedlings, and then decreased after 10 days of UV-B treatment in both transgenic lines compared to WT seedlings. (**C**) The metabolite profile of transgenic seedlings also exhibited more increased content of total flavonoid accumulation and maximized up to 4.76 mg/g after five days of UV-B treatment than the WT and then decreased after ten days UV-B treatment. The HPLC analysis also corroborated that transgenic lines showed higher flavonoid content than WT plants. One-way ANOVA was used to calculate statistical significance among different groups. The asterisks represent (*: *p* > 0.05, **: 0.01 < *p* < 0.05, ***: *p* < 0.01).

**Figure 9 molecules-28-03205-f009:**
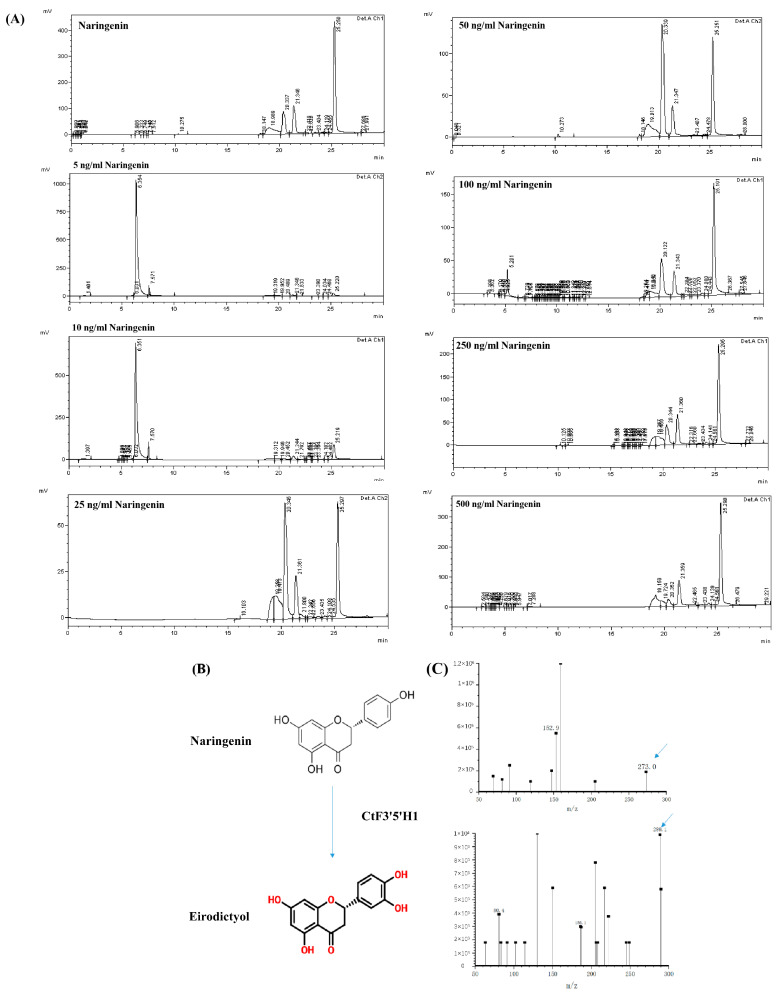
In vitro catalytic activity of recombinant CtF3′5′H1-encoding enzyme using HPLC-MS analysis. (**A**) HPLC analysis of different flavonoid substrates catalyzed by recombinant CtF3′5′H1-encoding enzyme. The intensity of product changes by CtF3′5′H1 was observed with the increase of substrate concentration. (**B**) The oxidation capacity of naringenin to eriodictyol catalyzed by recombinant CtF3′5′H1. (**C**) Time-course study of the enzymatic reaction, and LC/MS analysis of the substrate (naringenin) and product (eriodictyol). One-way ANOVA was used to calculate statistical significance among different groups.

**Figure 10 molecules-28-03205-f010:**
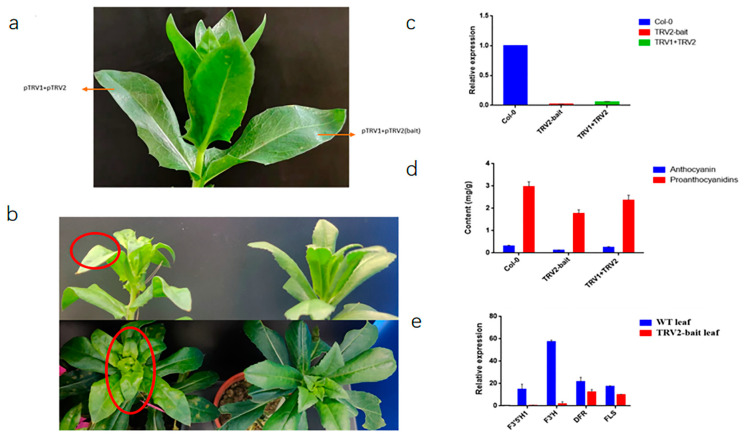
VIGS and difference analysis of phenotypic, relative expression, and metabolites accumulation in *CtF3′5′H1-*silenced safflower leaves. (**a**) The site of virus injection for VIGS analysis. (**b**) The phenotype of silenced safflower leaves exhibiting a light-yellow color and curly leaf than the non-silenced safflower leaves. The red circles denote safflower leaves at the period before the bud stage, and the mature leaves under the sepals, which were used for VIGS analysis. (**c**) Expression analysis of *CtF3′5′H1* in silenced and non-silenced safflower leaves. (Col-0: Wild type safflower without injection). (**d**) Quantification of anthocyanidin and pro-anthocyanidin accumulation in silenced and non-silenced safflower leaves. (**e**) The expression analysis of key genes involved in the downstream regulatory pathway of anthocyanin biosynthesis in silenced and non-silenced safflower leaves.

**Figure 11 molecules-28-03205-f011:**
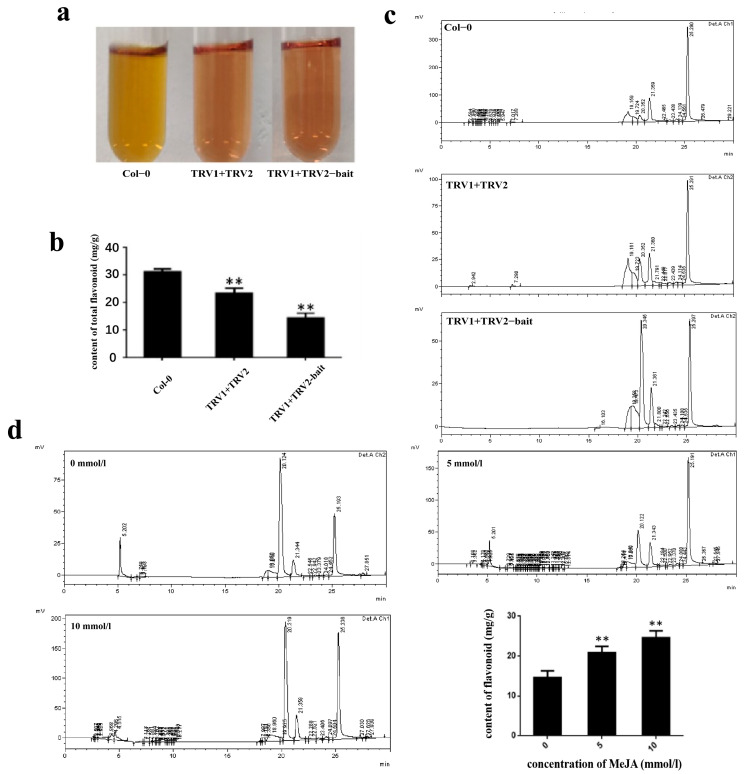
Quantification of total flavonoids content in safflower leaves under normal and MeJA treatment after VIGS analysis. (**a**) Color changes of metabolites extracts before and after silencing through VIGS. (**b**) Significant differences of total flavonoids content in silenced and non-silenced safflower leaves using HPLC-MS analysis. (**c**) The chromatographic separation of flavonoids content in silenced safflower leaves after different MeJA treatments. (**d**) Difference of total flavonoids content in silenced safflower leaves after different MeJA treatment. One-way ANOVA was used to calculate statistical significance among different groups. The asterisks represent (**: *p* > 0.01).

## Data Availability

All genes used in this study have been previously reported, and references are provided.

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
