# Peer review of "Safflower Flavonoid 3′5′Hydroxylase Promotes Methyl Jasmonate-Induced Anthocyanin Accumulation in Transgenic Plants"

_molecules, 2023, doi:10.3390/molecules28073205_

Round 1
Reviewer 1 Report
The article "Molecular Characterization of Safflower flavonoid 3'5'Hydroxylase and effects of Methyl Jasmonate on anthocyanin accumulation in transgenic Plants" screened two F3'5'H genes from online safflower transcriptome data and performed a bioinformatic analysis of them. The function of CtF3'5'H1 was also identified by means of Arabidopsis thaliana transgenic plants, methyl jasmonate stress and VIGS, and the gene was found to regulate total flavonoid, anthocyanin and proanthocyanidin synthesis. The study has some practical value, but there are still some problems, as follows:
1. We all know that the pigment-like components and main active ingredients of safflower belong to chalcones, and the authors mention that CtF3'5'H1 can alter the total flavonoid and proanthocyanidin content, but what is the effect of CtF3'5'H1 on the accumulation of chalcones like HSYA?
2. The authors used safflower leaves as material for the silencing of the CtF3'5'H1 gene by VIGS, while the main flavonoids of safflower are mainly enriched in the petals, why not use the petals of safflower as experimental material?
3. Why was Arabidopsis chosen for the transgene when VIGS is progressive in safflower? It is important to know that although Arabidopsis is the model plant, it is evolutionarily distantly related to safflower.
4. The authors started screening out CtF3'5'H1 and CtF3'5'H7, why did later only validate the gene function of CtF3'5'H1?
5. The article says that the leaves of the transgenic Arabidopsis are greener, but this difference may be due to different photographic conditions. We suggest using data to show the difference, such as the electronic eye can technique.
6. Why does it say “The remaining eight overexpressed transgenic lines transformed with CtF3'5'H1 except OE9 and OE10 did not show noticeable expression level during qRT-PCR analysis” again in rows 232-234 when all 10 transgenic OE strains are shown to be significantly different in gene expression in the b panel of Figure 5?
All the pictures in the article are not clear enough. There are also problems with the writing, e.g. “Arabidopsis thaliana” should be italicised, and all Latin names and gene names should be italicised. Please write the full names of the species appearing in Figure 1 in the figure notes. Is the “*” in Figure 5 an indication of significance? Please indicate this in all missing figure notes.
Reviewer 2 Report
In the study, the authors revealed the underlying molecular basis of a putative F3'5'H gene from safflower imparting MeJA-induced flavonoid accumulation. The constitutive expression of CtF3'5'H1 gene was verified at different flowering stages in safflower indicating their diverse transcriptional control through flower development. The manuscript is well designed and carried out. The obtained data can well support the conclusions. Therefore, it is recommended to be accepted after the following minor issues have been addressed.
1. The quality of some graphs needs to be improved.
2. Some methods require additional references.
Reviewer 3 Report
This manuscript entitled "Molecular Characterization of Safflower flavonoid 3'5'Hydroxylase and effects of Methyl Jasmonate on anthocyanin accumulation in transgenic Plants (Manuscript ID: molecules-2275049)" is describes a study that carried out genetic and functional analysis of the flavonoid 3'5' hydroxylase in safflower.
Although the molecular biology of safflower has not been fully analyzed, the authors have isolated the gene, analyzed it biochemically and even analyzed its function in vivo using heterologous overexpresser and VIGS, showing that it is an important gene involved in the biosynthesis of flavonoid species. The results presented by the authors in this paper should provide much insight into industrially important crops. However, there are some problems with this paper.
I think the data is very interesting but redundant overall. Also, there is a significant lack of explanation for each of the data.
Figures 6-8 need a clear explanation if the results are to be presented.
In particular, it is not at all clear where the differences are in the respective Arabidopsis photos. Each figure legend is also insufficient, and instead of showing only phenotypic changes, gene expression analysis, anthocyanin and proanthocyanidin content analysis, each result should be The reader cannot understand the results unless they are explained.
(Also, figures 7b and 8b appear to be the same data, but were they taken from different treatments?)
I would like to recommend the authors reconsider whether the MV and UV resistance results (Figs 7-8) are really necessary to demonstrate the significance of the study.
If they only want to show that CtF3'5'H were involved in increase flavonoids in Arabidopsis overexpressors, then only Fig. 5 is sufficient. On the other hand, if you want to show that F3'5'H gene expression is altered in response to stress, then analysis using native promoters may be necessary.
Minor points:
-The text in all figures is so small that it is completely unreadable. Please enlarge and reconsider the composition of the figures. Also, as mentioned above, the description of the figure legend is severely lacking. Please explain symbols and abbreviations.
It is strongly recommended that the manuscript is thoroughly checked by an English language editorial service specializing in biology (preferably plant molecular biology) before submission.
-Overall, there are inaccuracies in the notation. Scientific and gene names should be italicised.
-p.7, lines 243-244; there is an explanation of the protein electrophoresis picture in figure 5d, but there is no connection with the previous story. Also, if it is the result of Western blot, it is understandable, but if it is just the result of electrophoresis, I do not understand the meaning of showing it here.
-Figure 6b, 7b, 8b; I don't understand the meaning of "Meja intimidate", "MV intimidate". Also, I cannot find for which treatments the significance test results are for. Please provide an appropriate description.
-p.15, lines 362-365; I think this part should be listed under materials and methods.
-p.16, line 395-396; I think this part should be described in materials and methods. Several other similar minutes are found.
-figure 11a; In this picture, the colour of the extract seems to be darker with the VIGS treatment. However, the lack of description for the results makes it incomprehensible.
p.21, lines 605-621 (chap. 4.8); please elaborate on abiotic sress; there is no description of which MeJA or MV products were used, how the UV was treated (fluorescent lamp or LED, intensity, etc.) .
Round 2
Reviewer 1 Report
The author has made some changes to the article, but there are still unresolved concerns with the article. The specific problems are as follows:
1. The anthocyanin content of safflower is relatively small, so why did the authors choose to measure anthocyanins? In the safflower flavonoid metabolic pathway, the direct catalytic product of F3'5'H is not anthocyanin, so why not measure the content of the direct product? It is generally thought that this would give a more direct response to the function of the gene.
2. All pictures in the manuscript are still not as clear as required, especially the chromatograms. The chromatograms also do not identify the characteristic peaks, which are suggested to be in the supplementary material.
3. The authors used safflower leaves as experimental material for VIGS, but is the structure of safflower leaves suitable for infiltration and what is the efficiency of infiltration? It is suggested that the transformation efficiency should be reflected by fluorescence or staining. The effectiveness of this method cannot be reflected only by the relative gene expression and flavonoid content alone.
Reviewer 3 Report
I confirmed that the authors understood the reviewer comments and revised the manuscript.
However, there are still some errors and inadequate explanations that should be revised.
- Figure 6b, 7b and 8b: The results of the significance difference test are shown, but what does the "****" mean? Figure legend only explains * (p<0.05) and **(p<0.01).
Also, please clarify which treatment these are for comparison.
- Figure 8 and p.22 line 797: Ultraviolet-b and uv-b should be Ultraviolet-B and UV-B respectively.
Overall, the term UV and the terms Ultraviolet-b and uv-b are not unified.
- As mentioned in the Instructions for Authours, "SI Units (International System of Units) should be used".
The following is a basic rule: there should be space between numbers and units.
Also, in p.22 line 797: There is a mistake in the units (80w/cm, 100uw/cm2 should be "80 W/cm2, 100 μW/cm2").
Please review the notation again for the use of the units.
Minor points:
- p.3, line 98: does "SCFCOI1" mean "SCFCOI1"? Please confirm.
- p.6, line 218: "Nicotiana benthamiana" is a scientific name.
- p.13, line 449-450 :I think "m/z" should change to italic font.
- p.12, line 427-439: In Fugue 8a, which are the 5th days and which are the 10th days? Please describe them properly!
- Figure 9a is broken. Please fix it.
-p.18, line 666: "cytochrome -b5" is "cytochrome-b5" .
